# Delayed Quality Deterioration of Low-Moisture Cereal-Based Snack by Storing in an Active Filler-Embedded LDPE Zipper Bag

**DOI:** 10.3390/foods11121704

**Published:** 2022-06-10

**Authors:** Youngje Jo, Eunghee Kim, Sangoh Kim, Choongjin Ban, Seokwon Lim

**Affiliations:** 1Research and Development Department, B.E.T., Busan 48119, Korea; youngje0426@gmail.com; 2Center for Food and Bioconvergence, Seoul National University, Seoul 08826, Korea; dkrnd2@snu.ac.kr; 3Department of Plant and Food Sciences, Sangmyung University, Cheonan 31066, Korea; samkim@smu.ac.kr; 4Department of Environmental Horticulture, University of Seoul, Seoul 02504, Korea; 5Department of Food Science and Biotechnology, Gachon University, Seongnam 13120, Korea

**Keywords:** active zipper bag, cereal-based snacks, water vapor capability, microstructure

## Abstract

This study focused on controlling the vapor permeability of an active zipper bag and preserving the quality of cereal-based snacks during the storage period at home. The active zipper bag was prepared by extruding low-density polyethylene with active fillers obtained from natural mineral materials. The active zipper bag showed the same transparent appearance as the existing one but showed 21% lower water vapor capability. As a result, during a 20-day storage period, three types of grain-based snacks (biscuits, shortbread cookies, and puffed snacks) showed delayed increases in weight, moisture content, and moisture activity when stored in an active zipper bag. In addition, this also affected the texture of the biscuits and shortbread cookies, in which the area under the curve was reduced significantly after appearing at a peak during the hardness measurement. On the other hand, the decrease in the number of air cell fracture events in puffed snacks was remarkable. This result suggests that the inner microstructure is preserved better when stored in an active zipper bag. In conclusion, the active zipper bag showed poor water vapor permeability, suggesting that the prepared zipper bag can be developed as snack packaging.

## 1. Introduction

Cereal-based snacks, as a representative low-moisture food, exhibit relatively good stability for preservation compared to high-moisture foods because of the rare possibility of microorganisms’ growth and chemical reactions [1,2]. On the other hand, the ingredients comprising the inner microstructures of cereal-based snacks are exposed directly to air and either sorb or desorb moisture easily from or into the air as the humidity condition changes [3]. Changes in the moisture content of cereal-based snacks alter the water activity (*a*_w_), resulting in changes in the inner-structure properties (e.g., the glass state to the rubbery) [4]. The inner-structural changes induce the recrystallization of the ingredients, such as sugars and lipids, and accelerate the deterioration in texture, as the storage time is prolonged [5]. The degree of textural sensitivity that responds to changes in the moisture content differs according to the cereal-based snack, such as biscuits, cookies, and puffed snacks [6,7,8]. In addition, the increased *a*_w_ elevated the growth of microorganisms and increased the chemical reactions, such as lipid oxidation [9]. Therefore, cereal-based snacks having low moisture should be maintained under low-humidified conditions for storage to delay texture deterioration. In this respect, the appropriate utilization of some packaging films for storage can be an economical and effective tool to preserve the quality of cereal-based snacks.

A low-density polyethylene (LDPE) zipper bag is a packaging material commonly used in households for storing various types of food. On the other hand, because general LDPE packaging materials have relatively free moisture permeation, they are unsuitable for storing cereal-based snacks with significant changes in physical properties, even with a small amount of moisture absorption. Various efforts have been made to control the moisture permeation of the packaging material. One representative method is to disperse a nano-filler in the polymer matrix. Fillers composed mainly of metallic nano/microparticles or composites suppress moisture permeation by extending a path through which the moisture in the film diffuses. Therefore, various studies have been conducted to increase the storability of foods by suppressing water permeation using various types of inorganic fillers, such as AgNP, AuNP, ZnO, TiO_2_, and MgO [10,11,12,13,14].

Several active packaging studies using fillers have been conducted. Moreover, although cereal-based snacks are very important foods that require the suppression of moisture permeation during the storage period at home, changes in the storage properties of cereal-based snacks due to active packaging have not been studied. This study examined whether LDPE zipper bags with an active filler can increase the storability of cereal-based snacks at home. Using a zipper bag embedded with size-reduced active fillers, this study investigated how the inner microstructure and texture of the three cereal-based snack samples change over a certain storage period and compared them with the changes in the quality of the samples stored in a general LDPE zipper bag.

## 2. Materials and Methods

### 2.1. Materials

An active mineral filler powder (Anti-Rot 34; Korean patent number: 10-1494381), blended with and composed of loess, barley stone, jade, iron-coated zeolite, charcoal, kaolin, feldspar, biotite, hydroxyapatite, and selenium, was supplied by Han-green Tech Ltd. (Seoul, Korea). Commercial low-density-polyethylene (LDPE) zipper bags (width: 25 cm; height: 30 cm; Clean zipper bag, Clean wrap Co., Busan, Korea), with no fillers embedded, were purchased as typical zipper bags from a local market and used as a control. Cereal-based snacks, biscuit (46.0 g·per piece; composition: 60% *w*/*w* carbohydrates (6.7% *w*/*w* sugars), 6.7% *w*/*w* proteins, 30% *w*/*w* lipids (15.3% *w*/*w* saturated fats), and 0.5% *w*/*w* sodium; ingredients: wheat flour, shortening, blended edible oil, white sugar, glucose, whole milk powder, refined salt, processed milk cream, baking powder, etc.; Ace, Haitai Confectionery & Foods Co., Ltd., Seoul, Korea], shortbread cookies (72.7 g·per piece; composition: 72.6% *w*/*w* carbohydrates (28.6% *w*/*w* sugars), 5.4% *w*/*w* proteins, 20.2% *w*/*w* lipids (9.5% *w*/*w* saturated lipids), and 0.2% *w*/*w* sodium; ingredients: wheat flour, white sugar, starch, shortening, egg, processed butter, baking powder, refined salt, lecithin, cream extract, etc.; Sablé, Haitai Confectionery & Foods Co., Ltd., Seoul, Korea), and puffed snack (11.1 g·per piece; composition: 88% *w*/*w* carbohydrates (49.3% *w*/*w* sugars), 4% *w*/*w* proteins, 6.1% *w*/*w* lipids (2.3% *w*/*w* saturated lipids), and 0.1% *w*/*w* sodium; ingredients: corn mill, white sugar, rice bran oil, banana powder, palm oil, skim milk powder, refined salt, flavoring agents, egg white powder, and turmeric powder; Banana kick, Nongshim Co., Ltd., Seoul, Korea), were chosen as the low-moisture food samples and were obtained from the local market. LDPE was purchased from LG Chem Ltd. (Seoul, Korea). All other chemicals were of analytical reagent grade.

### 2.2. Preparation of the Active Zipper Bag

The active mineral filler powder was ground using a jet-mill system (4″ Micronizer, Sturtevant Inc., Hanover, MA, USA) to reduce the particle size and size distribution. Constant feeding (0.55 MPa) and grinding (72 MPa) pressure values were used, and a solid feeding rate was maintained in a range of 0.45–0.91 kg·h^−1^ using a volumetric feeder (Schenck AccuRate Inc., Whitewater, WI, USA). The active filler ground was mixed well with LDPE of three-times the weight of the filler and placed into an extruder chamber (180 °C) to prepare the pellet (diameter: 2.5–3 mm). The pellet was then compressed and extruded as the film (thickness: 40–45 μm) using a roller. The zipper bags (width: 25 cm; height: 30.5 cm) were then prepared with the film using a continuous band heat sealer.

### 2.3. Particle Size Measurement

The volumetric particle size of the active mineral filler ground was measured using a laser diffraction analyzer (S3500, Microtrac Inc., Montgomeryville, PA, USA). For the measurement, a hundred milligrams of the filler ground powder was dispersed in a dry dispersion system (2 bar, 25 °C). The volumetric mean diameter was reported as follows:(1)$$\mathrm{MV}=\sum V_{i}d_{i}/\sum V_{i}$$
where $V_{i}$ is the volume percent of the particles of diameter $d_{i}$ in the population, which was calculated based on the volumetric size distribution measured.

### 2.4. Scanning Electron Microscopy

Field-emission scanning electron microscopy (FE-SEM; AURIGA, Carl Zeiss, Jena, Germany) of the ground filler and filler-embedded film was performed to ensure the reliability of MV and examine the filler distribution in the film matrix. In addition, micrographs of low-moisture model foods were also obtained to examine the microstructures.

### 2.5. Water Vapor Permeability Test

The permeation rate (permeability) of water vapor through the film was measured by cutting zipper bags as circular films with a diameter of 7.7 cm. The film cuts were placed under constant conditions (temperature: 25 °C; relative humidity (RH): 50%) 24 h before the measurements. A water vapor permeability tester (vapometer) was used to measure the film permeability of 135 mL double-deionized water (DDW). The weight of the tester stored in the center of the chamber was recorded 24 h after storage. The mass flux (g·m^−2^·h^−1^) of film-permeating water vapor was calculated as follows:(2)$$\mathrm{permeability}=\frac{W_{t}-W_{0}}{A_{f}\cdot t}$$
where $W_{t}$ and $W_{0}$ are the weights of the tester after and before the storage of 24 h, respectively, $A_{f}$ is the film area exposed to the atmosphere, and t is 24 h for storage. All measurements were conducted at 25 °C and 50 RH% using a thermos-hygrostat chamber (LHT-0100M, Daihan Labtech Co. Ltd., Namyangju, Korea).

### 2.6. Storage for the Cereal-Based Snacks

Three cereal-based snacks, biscuit, shortbread cookie, and puffed snack, were used as the low-moisture model foods. Immediately after unpacking from the original package, 10, 10, and 20 pieces of biscuit, shortbread cookie, and puffed snack, respectively, were placed in zipper bags. Each bag was stored for a predetermined time frame (either 1, 5, 10, 15, or 20 days) at 25 °C and 50 RH%. The sample snacks in the individual zipper bag were used only for one of the following measurements: total weight, moisture content, *a*_w_, and texture, at a storage period of 1, 5, 10, 15, or 20 days.

### 2.7. Assessment of Total Weight

The weight difference of all the zipper-bag-stored snacks from the initial weight was measured at each predetermined time (1, 5, 10, 15, and 20 days) while being stored in the bags. At least three independent zipper bags containing the snacks were weighed to determine the weight changes during storage once. In addition, at least three independent storage tests for the snacks in the zipper bags were performed to calculate the means and standard deviations of these experiments.

### 2.8. Analyses of Moisture Content, Water Activity, and Texture

The moisture content of the snacks stored in the zipper bags was measured using an infrared light moisture analyzer (MB45, Ohaus Inc., Pine Brook, NJ, USA) at a drying temperature of 105 °C. The *a*_w_ of the stored snacks was determined using an AquaLab 4TE chilled-mirror dew-point water activity meter (METER Group Inc., Pullman, WA, USA). The force deformation curves were obtained using a texture analyzer (TA.XT Express, Stable Micro Systems Ltd., Godalming, Surrey, UK), with a three-point bending rig at 0.6 mm·s^−1^. Force and time data were recorded with Texture Expert (Version 1.0) software from Stable Micro Systems. In a single measurement, one piece of the snacks was selected randomly from the zipper bags to measure the moisture content (%), *a*_w_, or texture. The moisture content, *a*_w_, and texture of at least three pieces in a bag were measured individually. In addition, the snacks incorporated in one of the zipper bags stored were used independently for “the at least three measurements” at one of the predetermined storage days and were not used for other measurements.

### 2.9. Statistical Analysis

Statistical analysis included an analysis of the variance with a *t*-test (Shapiro–Wilk test, equal variance test, and Mann–Whitney rank-sum test), performed using SigmaPlot version 12.0 (IBM Corp., Armonk, NY, USA). Data are presented as the means of at least three independent experiments. A *p* < 0.05 was considered statistically significant.

## 3. Results and Discussion

### 3.1. Characteristics of the Active Zipper Bag

The MV of the active mineral filler ground used in this study was ~3 μm. A unimodal peak size was obtained, as shown in the volumetric size distribution (Figure 1A). In addition, the size data were also verified from the micromorphology in the micrograph. Therefore, the active filler particles can be embedded well in the matrix of the LDPE film unless they aggregate during the extrusion procedures. No filler particle aggregates were observed in a micrograph of the film (Figure 1B). This result suggests that the ground active mineral filler was stably dispersed, even though a large amount (2.5% *w*/*w*) was used to prepare the active film. The extrusion procedures were applied twice. The active film zipper bag prepared with this film was less transparent than the typical LDPE zipper bag (Figure 1C,D), which is due to the large size of the embedded fillers (MV: ~3 μm) scattering light.

The calculated water vapor permeability of the active zipper bag was 5.60 × 10^−5^ g·m^−2^·h^−1^, which was ~21% lower than that of a typical zipper bag (7.06 × 10^−5^ g·m^−2^·h^−1^). This suggests that a smaller amount of moisture can permeate and diffuse into the inner space of the zipper bag from the outside, and vice versa. Furthermore, based on various mechanical characterizing tests guaranteed by the Korean Agency for Technology and Standards (KATS) (KS M 3001, testing method for mechanical characteristics of polyethylene film, 2016; International Classification for Standard (ICS) code: 83.140 and 83.080.20), the tensile strength, tensile stretch (breaking elongation %), and internal tearing strength of the prepared active zipper bag were 2.79 kN·cm^−2^, 645%, and 1.29 kN·cm^−1^, respectively. This result indicates that the active zipper bag exhibits similar mechanical properties to typical commercial zipper bags.

Two main things affect the moisture permeability of the film: the complexity of the moisture permeation path, expressed as tortuosity, and the hydrophilicity of the film components that allows moisture to remain in the film [15]. In both active and general zipper bags, the main material was LDPE; the filler content of the active zipper bag was only 2.5% *w*/*w*. Hence, the hydrophilicity of the material has a minimal effect on moisture permeability. Consequently, the lower water permeability of the active zipper bag has been attributed to the tortuous pathway generated by the embedded filler particles. In some studies [12,16], the particles embedded in the film might have affected the narrowing of the pore channel and the extension of the diffusion path.

### 3.2. Total Weight of the Cereal-Based Snacks during Storage

The changes in the weight of the cereal-based snacks (Δ weight) were monitored during the storage time. All the snacks became heavier for 20 days, regardless of the packaging bags (Figure 2). A detailed Δ weight was examined during the storage of biscuits in typical and filler-embedded (active) LDPE zipper bags for 20 days; the Δ weight increased to ~0.15 g and ~0.13 g, respectively. At each predetermined time (1, 5, 10, 15, or 20 days), the Δ weight of the biscuits in the typical bags was larger than those in the active bags, but there was no statistically significant difference (Figure 2A). In addition, for both the storage zipper bags, the increasing rate of the Δ weight decreased with increasing storage time, suggesting the intermediate state to Δ weight saturation. For the Δ weight of shortbread cookies stored in the typical and active bags, the Δ weight increased to ~0.35 g and ~0.26 g, respectively, with a slight decrease in the increasing rate during storage (Figure 2B). Twenty days after initiating storage, a significant difference in Δ weight was observed between the cookies stored in the typical and active bags. For the Δ weight of puffed snacks stored in typical and active bags, the Δ weight increased to ~0.05 g and ~0.04 g, respectively (Figure 2C). At every time monitored, all the Δ weight of the puffed snacks in the typical bags was higher than those in the active bags, but there was no statistically significant difference. For the snacks in the typical and active bags, the Δ weight increased rapidly during the initial 10 days of storage, but little change was observed after that time, indicating saturation.

### 3.3. Moisture Content and Water Activity of the Cereal-Based Snacks during Storage

The changes in Δ weight of the low-moisture cereal-based snacks during storage were mainly attributed to the moisture sorption of the ingredients constituting the snack matrix, such as various proteins and carbohydrates. Accordingly, the changes in the moisture content (% dry total) of all the cereal-based snacks were recorded during storage (Figure 3). The moisture content was measured during the storage of biscuits in typical and active LDPE zipper bags for 20 days; the moisture content increased from ~1.6% to ~3.6% and ~3.4%, respectively (Figure 3A). At each predetermined time, there was no significant difference in the moisture contents between the typical and active bags. Similar to the trend in the Δ weight (Figure 2A), the increasing rate of the moisture contents decreased with time, regardless of the storage zipper bag type. For the shortbread cookies stored in the typical and active bags, the moisture contents increased from ~1.0% to ~3.4% and ~3.0%, respectively, with a slight decrease in the growth rate of the moisture contents during storage (Figure 3B). In particular, five days after storage initiation, a significant difference in moisture content was observed between the cookies stored in the typical and active bags, which probably influenced the Δ weight difference between the typical and active bags after storage. For the puffed snacks stored in the typical and active bags, the moisture contents increased from ~2.1% to ~4.9% and ~4.8%, respectively (Figure 3C). No significant difference in the moisture content between the typical and active bags was observed during all storage times. Furthermore, similar to the tendency in the Δ weight (Figure 2C), saturation of the moisture contents was also observed 10 days after the storage initiation. Consequently, the changes in the moisture content of the stored cereal-based snacks critically affect the weight, and shortbread cookies were the most sensitive to the storage bag type. For the storage of all sample snacks, the same zipper bags, typical and active, having a moisture permeability of 5.60 × 10^−5^ and 7.06 × 10^−5^ g·m^−2^·h^−1^, respectively, were used. Only the shortbread cookies showed a significant difference in the moisture content between the typical and active bags. This result suggests that the largest moisture-sorbing capacity of shortbread cookies might be due to the differences in ingredients, production process, structure, and other factors among the samples.

The *a*_w_ of foods does not have a linear relationship with the water content, but the relationship follows a curve known as a moisture sorption isotherm [17,18]. Monitoring *a*_w_ during storage makes the water migration predictable between foods and the surrounding conditions [19,20]. In addition, ingredients and composition for specific foods affect the *a*_w_ having an isotherm relationship with the moisture content [21,22]. The ingredients and composition of specific low-moisture foods are rarely changed during short-term storage, but the water sorption can induce changes in the structure and influence the *a*_w_ [23]. Therefore, monitoring *a*_w_ during the storage also allows for indirect predictions of the structural changes in the inner matrix of foods. Accordingly, the changes in the *a*_w_ of the cereal-based snacks were also monitored to predict the water migration and inner matrix changes (Figure 4). For the changes in the *a*_w_ for 20 days, the *a*_w_ of biscuits stored in the typical and active bags increased from ~0.10 to ~0.40 and ~0.37, respectively (Figure 4A). At each predetermined time, no significant differences in the moisture content were observed between the typical and active bags. Similar to the trend in the water content (Figure 3A), the increasing rate of the *a*_w_ values decreased with time, irrespective of the storage zipper bag type. This suggests moisture sorption without ingredient rearrangement or structural changes. The *a*_w_ of the shortbread cookies stored in the typical and active bags increased from ~0.10 to ~0.35 and ~0.35, respectively, with a slight decrease in the increasing rate of *a*_w_ values five days after storage (Figure 4B). In addition, after storage for one day and five days, the *a*_w_ values were significantly different in the cookies stored in the typical and active bags. This tendency was similar to that of the water content (Figure 3B), suggesting moisture sorption without ingredient rearrangement and structural changes, similar to that of biscuits. The *a*_w_ of puffed snacks stored in the typical and active bags increased from ~0.11 to ~0.44 and ~0.44, respectively (Figure 4C). No significant differences in the *a*_w_ were observed between the typical and active bags during all storage times. On the other hand, in contrast to the trends in the Δ weight (Figure 2C) and water content (Figure 3C), the saturated *a*_w_ values increased from 5 and 10 days of storage in the typical and active bags. This result indicates ingredient rearrangement and simultaneous inner-structural changes caused by moisture sorption, which is in contrast to that observed for biscuits and shortbread cookies. This might be because the puffed snack has a relatively weak inner structure and a large surface area, owing to its porous structure [24]. Overall, the changes in the *a*_w_ of the stored cereal-based snacks affected the moisture content non-linearly, and the puffed snacks probably underwent ingredient rearrangement and inner-structural changes 15 days after storage.

### 3.4. Texture and Microstructure of the Cereal-Based Snacks during Storage

Increases in moisture content and *a*_w_ of the low-moisture snacks alter the inner microstructure, and eventually, worsen the texture and mouthfeel [25]. In this regard, the force deformation curves of the cereal-based snacks in either typical or active zipper bags were obtained during storage, and the changes in the texture properties were analyzed (Figure 5). In the deformation curves of the biscuits and shortbread cookies, a unimodal peak and force-decreased profile were observed after the peak appeared, as shown in Figure 5A. The force at the peak is defined as the hardness [26]. The texture profiles before and after the peak reflect the texture properties before and after transverse rupture during the measurements, respectively. In this study, the integral area under the profile curve after the peak is defined as Area 2. For biscuits, a lower hardness or larger Area 2 indicates lower crunchiness or higher sogginess. The values for hardness (Figure 5B) of both biscuits stored in the typical (~10.0 to ~6.1 N) and active (~10.0 to ~6.6 N) bags were decreased as storage time passed. The values for Area 2 (Figure 5C) of both biscuits stored in the typical (~5.2 to ~11.9 N·s) and active (~5.2 to ~6.9 N·s) bags were increased as storage time passed. This result indicates decreased crunchiness or increased sogginess due to moisture sorption. In particular, during storage, the Area 2 values of the biscuits in the active bags increased at a significantly lower rate than those in the typical bag, whereas there was no significant difference in the hardness. Hence, the active zipper bag had a superior effect in maintaining the texture of biscuits for 20 days compared with the typical bags. For shortbread cookies, in which the softness is important, the moisture sorption and the increased *a*_w_ altered the water distribution and hardened them due to sucrose recrystallization [27]. Unlike the case for biscuits, the hardness (Figure 5D; typical: ~9.0 to ~14.9 N; active: ~9.0 to ~12.8 N) and Area 2 (Figure 5E; typical: ~4.5 to ~17.6 N·s; active: ~4.5 to ~9.9 N·s) increase over the 20-day storage period, irrespective of the storage zipper bag type. A significant difference between the typical and active zipper bags was observed clearly in Area 2. This result suggests that the active zipper bag helps maintain the softness of the cookies for a more extended period than the typical bags.

Puffed snacks becoming moisturized undergo textural softening and a reduction in crispiness due to moisture migration into the air cell structures, the water dissolution of the ingredients, and the glass transition [28,29,30]. Many sharp peaks were observed in the force deformation curves of non-stored puffed snacks (Figure 5F; positive case). A larger number of the peaks appearing in the deformation curves indicate a better mouthfeel because an individual peak reflects the fracture of an individual inner microstructure [28]. The hardness of the samples stored in typical zipper bags increased from ~7.7 N to ~12.7 N after five days and then decreased to ~8.2 N after 15 days, while those for active zipper bags gradually increased to ~13.9 N (Figure 5G). As a result, the hardness of the samples stored in the active zipper bags was significantly greater than those stored in the typical zipper bags for 15 days. The number of air cell fracture events is expressed as crispness, which represents a major part of the quality of puffed snacks with multi-cell structures [31]. Based on the number of the air cell fracture events in the force deformation curves (*n*_f_; Figure 5H), *n*_f_ decreased with increasing storage time regardless of the storage zipper bag type (typical: ~43 to ~14; active: ~43 to ~18), indicating a loss of crispiness. Puffed snacks stored in active zipper bags had a higher *n*_f_ than those in the typical zipper bags, indicating better crispiness. Moreover, typical bag-stored samples exhibited only a few peaks for the brittle break event rather than many peaks for air cell fracture events five days after the storage, which was different from the active-bag-stored samples. This characteristic discrepancy in the deformation curves between the typical and active-bag-stored samples became more apparent as the storage time increased. After 20 days of storage, no peak for air cell fracture and brittle break events was observed in the deformation curves of typical-bag-stored samples (Figure 5I). On the other hand, the active-bag-stored samples still had several peaks for the air cell fracture and brittle breakage events. Hence, the active filler-embedded zipper bags effectively reduce the moisturization and texture deterioration of puffed snacks during storage.

The microstructures of all the cereal-based snacks, having the appearances as shown in Figure 6A, were investigated by FE-SEM. As observed in the microstructure of biscuits before storage (Figure 6B), many granules were immersed in a smooth matrix. The granules would be starch granules, and the continuous matrix would be formed by denatured proteins and smoothened by a melted fat coating on the surface [32]. The starch granules were probably not gelatinized because of the low water content in the formulations and water competition with proteins, sugars, and other ingredients. After storage in a typical zipper bag for 20 days (Figure 6C), compared to those before storage, the surface of the matrix became smoother and starch granules immersed in the matrix were less protruded from the surface. This result might be due to the recrystallization of the ingredients, such as sugars and lipids during storage, resulting from increased *a*_w_. Eventually, this could influence the decreased hardness and increased Area 2 (Figure 5B,C).

For shortbread cookies before storage, the microstructure, i.e., pores, was similar to that of biscuits (Figure 6B). Fat crystals become enveloped in a protein membrane during the dough-mixing procedure. This membrane allows a large number of crystals to attach to air bubbles. During baking, the fat crystals melt and the protein membrane is incorporated into the surface of the bubbles as they expand [33,34]. The microstructure, including the pores, results from the formation of the bubbles and the rearrangement of ingredients. After storage (Figure 6C), the pores shrink, possibly due to the increased *a*_w_ and the recrystallization of the ingredients, such as biscuits, during the storage. These changes increased only Area 2 of the biscuits, whereas the cookies showed an increase in both hardness and Area 2 (Figure 5D,E).

For puffed snacks, air cell structures were observed in the micrograph (Figure 6B), which were constructed from bubbles surrounded by gelatinized starch–lipid complexes and moisture evaporation [35]. After storage (Figure 6C), the wall comprising the gelatinized starch–lipid complexes and surrounding the air cells thickened compared to that before storage, possibly due to the swelling and rearrangement of the matrix by the increased *a*_w_. Because of the rubbery state of the matrix, deterioration can be observed during the storage in the textural characteristics, including hardness, *n*_f_, and brittle breaks in the deformation curve. Consequently, all the deteriorations of the cereal-based snacks during storage may have originated from moisture sorption. This may have had the effect of delaying the changes in the microstructure of cereal-based snacks during storage (Figure 6B,C). Moreover, storage in the active filler-embedded zipper bag effectively delays deterioration compared to a typical zipper bag.

## 4. Conclusions

The water vapor permeability of the active filler-embedded LDPE zipper bag used in this study was smaller than that of a typical zipper bag, which effectively preserved the quality of low-moisture cereal-based snacks during storage. Among the three low-moisture snacks, the moisture content and *a*_w_ of shortbread cookies increased more slowly when stored in the active zipper bag than in a typical zipper bag. Furthermore, the texture of all three low-moisture snacks stored in active zipper bags was better than that stored in typical zipper bags. Based on the microstructure of the low-moisture snacks, the better-preserved texture of the active-zipper-bag-stored snacks might be due to the lesser reconstitution, rearrangement, and recrystallization of the ingredients, resulting from the lower water sorption. Consequently, an active zipper bag to store low-moisture cereal-based snacks can improve the quality for at least 20 days compared to a typical zipper bag. In conclusion, this study provides fundamental knowledge to develop packaging materials for storing low-moisture foods without deteriorating the quality at home.

## Figures and Tables

**Figure 1 foods-11-01704-f001:**
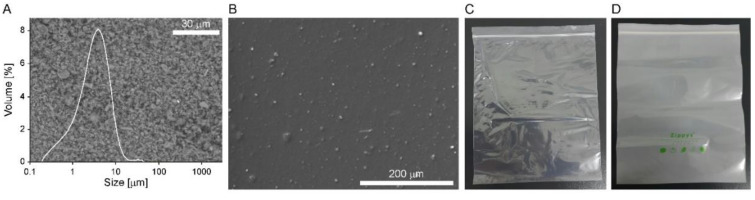
(**A**) Volumetric size distribution and scanning electron micrograph of the active mineral fillers ground. (**B**) Scanning electron micrograph of the film embedded with the fillers. Appearances of (**C**) a typical LDPE zipper bag and (**D**) zipper bag prepared with the active filler-embedded LDPE film.

**Figure 2 foods-11-01704-f002:**
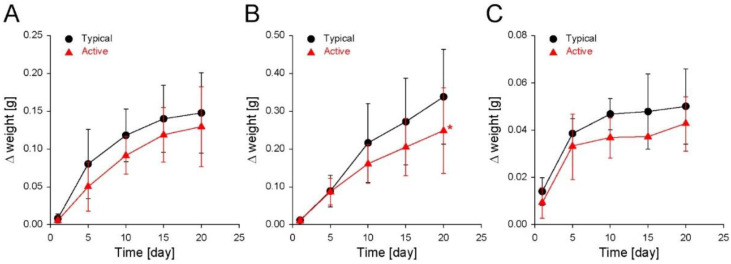
Changes in the weight (Δ weight) of (**A**) biscuits, (**B**) shortbread cookies, and (**C**) puffed snacks stored in either a typical LDPE zipper bag or an active filler-embedded LDPE zipper bag for 20 days (25 °C, 50 RH%). The data are expressed as the mean ± s.d. (standard deviation; *: *p* < 0.05).

**Figure 3 foods-11-01704-f003:**
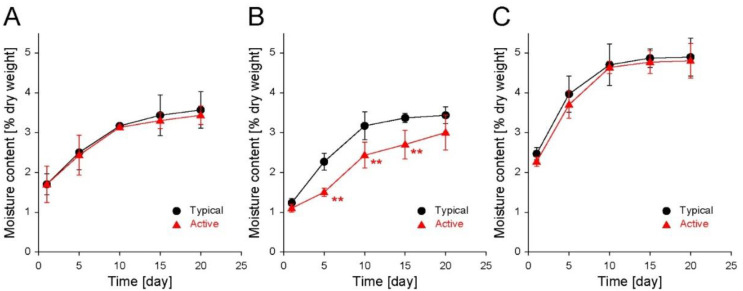
Changes in the moisture content of (**A**) biscuits, (**B**) shortbread cookies, and (**C**) puffed snacks stored in either a typical LDPE zipper bag or an active filler-embedded LDPE zipper bag for 20 days (25 °C, 50 RH%). The data are expressed as the mean ± s.d. (standard deviation; **: *p* < 0.01).

**Figure 4 foods-11-01704-f004:**
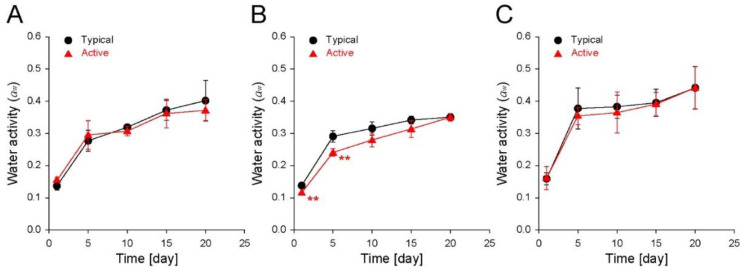
Changes in the water activity (*a*_w_) of (**A**) biscuits, (**B**) shortbread cookies, and (**C**) puffed snacks stored in either a typical LDPE zipper bag or an active filler-embedded LDPE zipper bag for 20 days (25 °C, 50 RH%). The data are expressed as the mean ± s.d. (standard deviation; **: *p* < 0.01).

**Figure 5 foods-11-01704-f005:**
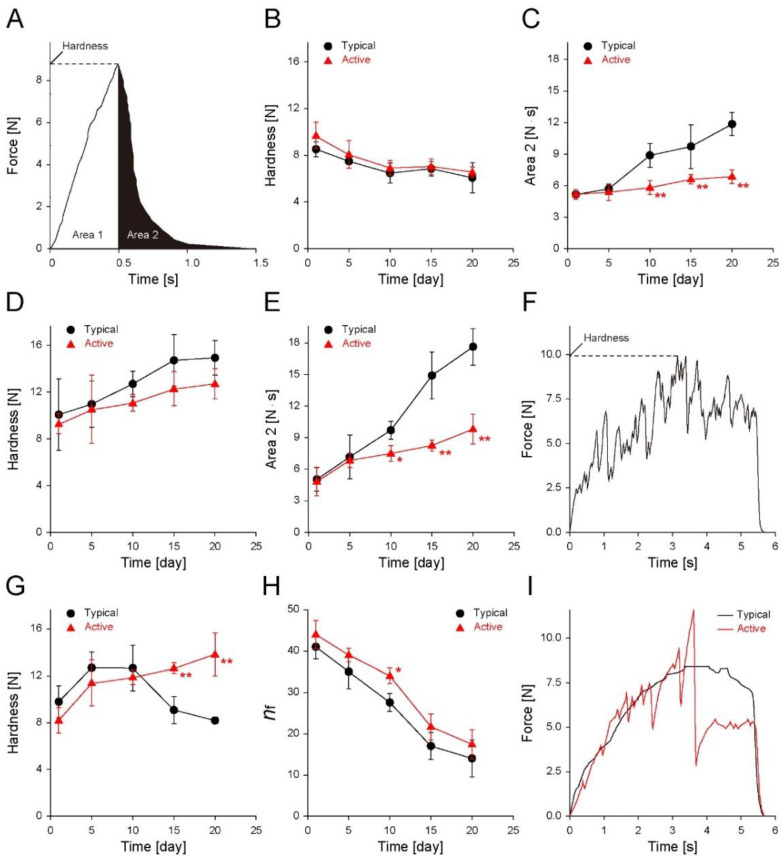
(**A**) Representative force deformation curve in texture analyses of biscuits and shortbread cookies (Area 1: area under the curve before a peak appeared; Area 2: area under the curve after showing a peak). Changes in (**B**) hardness and (**C**) Area 2 of biscuits stored in either a typical LDPE zipper bag or an active filler-embedded LDPE zipper bag for 20 days (25 °C, 50 RH%); changes in (**D**) hardness and (**E**) Area 2 of shortbread cookies during storage. (**F**) Representative force deformation curve in texture analyses of native puffed snacks. Changes in (**G**) hardness and (**H**) *n*_f_ (the number of air cell fracture events) of puffed snacks during storage. (**I**) Representative force deformation curve in texture analyses of puffed snacks after 20 days of storage in either typical or the active zipper bag. The data are expressed as the mean ± s.d. (standard deviation; *: *p* < 0.05 and **: *p* < 0.01).

**Figure 6 foods-11-01704-f006:**
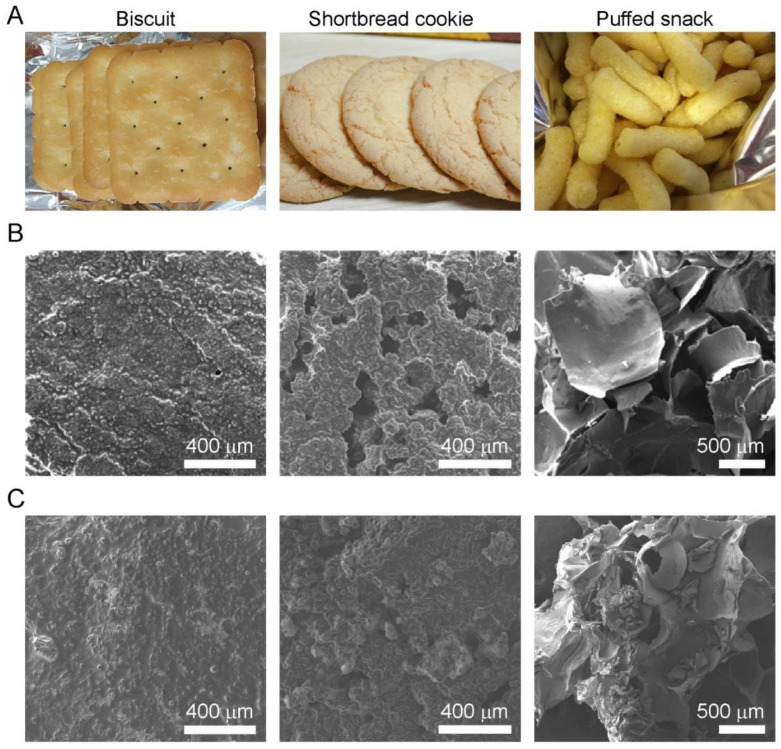
(**A**) Appearances of the cereal-based snacks (biscuit, shortbread cookie, and puffed snack). Scanning electron micrographs of the cereal-based snacks (**B**) before and (**C**) after storage in a typical LDPE zipper bag for 20 days (25 °C, 50 RH%).

## Data Availability

Data is contained within the article.

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
