# Peer review of "Delayed Quality Deterioration of Low-Moisture Cereal-Based Snack by Storing in an Active Filler-Embedded LDPE Zipper Bag"

_foods, 2022, doi:10.3390/foods11121704_

Round 1
Reviewer 1 Report
Comments are provided in the attached document.

Author Response
The paper is very interesting, well written and various for reading. The presentation of the results is clear, the discussion is good.
Point 1: Line 35: Delete the dot in front of the parentheses [3].
Response 1: We have deleted as the reviewer mentioned.
Point 2: Paragraph 2.8: Can you give the exact name of the method you used within the Texture Analyzer Software?
Response 2: We have added the information as the reviewer requested (lines 147-148).
Point 3: Paragraph 3.4: You should not use these force units (kgf), it does not comply with the International System of Units (SI), use newton (N).
Response 3: We have revised “kgf” to “N” as the reviewer requested.
Point 4: Lines 301-303: Write this sentence a little clearer.
Response 4: We have separated this sentence to write clearer, as the reviewer mentioned.
Point 5: Figure 6: It would be useful to present scanning electron micrographs of cereal based snack storage in active zipper bag.
Response 5: Unfortunately, it is difficult to present scanning electron micrographs of cereal-based snack storage in active filler-embedded zipper bag for 20 days because the time given for revision is only 10 days. However, it can be inferred that the samples stored in the active zipper bag absorbed less moisture than those stored in the typical zipper bag, and the deterioration was delayed accordingly, so the microstructure change was also small. Therefore, a related discussion has been added (lines 391-393).

Reviewer 2 Report
Dear Authors,
The manuscript presented for review, "Delayed quality deterioration of low-moisture cereal-based snack by stored in the active filler-embedded LDPE zipper bag", contains interesting results of a comprehensive study. The obtained results, apart from the scientific knowledge in the field of materials used in the production of active food packaging, are of great practical importance. They fit into the current research trends regarding the search for methods of extending the shelf life of food. The chapter "Introduction" is a good introduction to the subject of the work and justifies the purposefulness of the research undertaken. The material and research methods were correctly selected for the intended purpose of the study. The research results were statistically processed, correctly described, and discussed with the available literature. It is worth emphasizing that the graphical form of presenting the obtained results is very clear. The conclusions are fully justified by the obtained results.
Notes for work:
Line 35 - please delete the character.
Line 116 - please delete the character;
Line 183 - unclear what 645% refers to
References - DOI numbers are missing, please complete
Author Response
The manuscript presented for review, "Delayed quality deterioration of low-moisture cereal-based snack by stored in the active filler-embedded LDPE zipper bag", contains interesting results of a comprehensive study. The obtained results, apart from the scientific knowledge in the field of materials used in the production of active food packaging, are of great practical importance. They fit into the current research trends regarding the search for methods of extending the shelf life of food. The chapter "Introduction" is a good introduction to the subject of the work and justifies the purposefulness of the research undertaken. The material and research methods were correctly selected for the intended purpose of the study. The research results were statistically processed, correctly described, and discussed with the available literature. It is worth emphasizing that the graphical form of presenting the obtained results is very clear. The conclusions are fully justified by the obtained results.
Notes for work:
Point 1: Line 35 - please delete the character.
Response 1: We have deleted as the reviewer mentioned.
Point 2: Line 116 - please delete the character;
Response 2: We have deleted as the reviewer mentioned.
Point 3: Line 183 - unclear what 645% refers to
Response 3: “645%” referred “tensile stretch”. To clarify it, we have added “(breaking elongation %)” after “tensile stretch” in a line 184.
Point 4: References - DOI numbers are missing, please complete
Response 4: We have added DOI numbers as the reviewer requested.

Reviewer 3 Report
The article deals with the impact of the active zipper bag on the quality of cereal-based snacks during storage. The article is very interesting and important for the development of new packaging for snacks
Author Response
The article deals with the impact of the active zipper bag on the quality of cereal-based snacks during storage. The publication is very interesting and important for the development of new packaging for snacks.
Response 1: We greatly appreciate the reviewer’s positive evaluation.

Reviewer 4 Report
Authors Jo et al. investigated the quality parameters of cereal snack during 20 days of storage in LDPE packaging with or without active filler. The work is interesting; however, the authors need to explain why the storage time was so short, since these products typically have a shelf-life of 180 days. The short storage is probably the reason why differences in investigated quality parameters were small or insignificant. Another limitation of the study is that oxygen permeability was not assessed, whereas the oxidative instability of cookies/biscuits is the main reason of their spoilage. In addition, authors should provide a reference or explanation for the texture interpretation - usually brittleness, fracturability, toughness or work of failure are calculated along with the hardness.
Author Response
Authors Jo et al. investigated the quality parameters of cereal snack during 20 days of storage in LDPE packaging with or without active filler.
Point 1: The work is interesting; however, the authors need to explain why the storage time was so short, since these products typically have a shelf-life of 180 days. The short storage is probably the reason why differences in investigated quality parameters were small or insignificant.
Response 1: As far as we know, the shelf-life of 180 days of low-moisture cereal-based snacks on shelfs of market is because of the water and oxygen block by the inner metal layer laminated in a packaging film bag and because of the nitrogen filling. This study is for packaging the snacks during storage after the purchase at home rather than for packaging the snacks in market. To clarify the purpose of this study, we have added the word “at home” (lines 60 and 63).
Point 2: Another limitation of the study is that oxygen permeability was not assessed, whereas the oxidative instability of cookies/biscuits is the main reason of their spoilage.
Response 2: This study is for delaying the texture deterioration of low-moisture snacks during storage. With this respect, we just focused on showing quality changes in the texture rather than others including the oxidative instability. To clarify the purpose of this study, we have added the word “at home” (lines 60 and 63).
Point 3: In addition, authors should provide a reference or explanation for the texture interpretation - usually brittleness, fracturability, toughness or work of failure are calculated along with the hardness.
Response 3: The parameters used in this paper were preferentially selected from those considered suitable to explain the quality change of cereal-based snacks. For example, the toughness was translated by the total area under the curve of stress versus strain (N/mm) and denoted the energy required to bite or chew the product. The toughness can be converted from the sum of area 1 and 2 (N·s) in Figure 5A by multiplying the test speed (0.6 mm·s–1) and then divided with sample height and probe compression area. This is because the test speed is the same in all measurements and differences in the sample height and probe compression area are negligible. It is known that the toughness can be increased due to the recrystallization of starch. However, in this study, the change in area 2 was more dramatic than the toughness covering the entire area 1 and 2. In addition, the large size of area 2 represents the high sogginess of the snack, and it better represents the quality of biscuit and shortbread cookies than toughness, so we paid attention to this and explained. Also, in the case of puffed snacks, the presence of air cells is a significant quality factor. Therefore, the quality of puffed snacks was expressed by changes in the number of air cell fracture events. Singh et al. took the same approach, defined the number of significant positive peaks as crispness, and explained the quality of crispy extruded snacks. To clarify the purpose of this study, we have added the explanations and citations (lines 330-331 and Ref. 31).

Round 2
Reviewer 4 Report
Authors Jo et al. have clarfieid the aim of their study which was essential. I suggest to add the purpose of the packaging, i.e. for home use also in the abstract and conclusion.
Author Response
Authors Jo et al. have clarfieid the aim of their study which was essential.
Point 1: I suggest to add the purpose of the packaging, i.e. for home use also in the abstract and conclusion.
Response 1: As the reviewer requested, we have added the word “at home” in the abstract and conclusion (lines 15 and 408).
